# Morphofunctional Alteration of Mesenteric Lymph Nodes in the Inflammation of the Abdominal Cavity

**DOI:** 10.3390/biology13030166

**Published:** 2024-03-04

**Authors:** Serik N. Abdreshov, Georgii A. Demchenko, Anar N. Yeshmukhanbet, Makpal A. Yessenova, Sandugash A. Mankibaeva, Gulshat K. Atanbaeva, Marzhan S. Kulbayeva

**Affiliations:** 1Laboratory of Physiology Lymphatic System, Institute of Genetics and Physiology SC MSHE RK, Almaty 050060, Kazakhstan; georgiidemchenko@mail.ru (G.A.D.); eshmukhanbet96@mail.ru (A.N.Y.); esenova_makpal@mail.ru (M.A.Y.); 2Faculty of Biology and Biotechnology, Al-Farabi Kazakh National University, Almaty 050040, Kazakhstan; mankibaevasandugash@mail.ru (S.A.M.); gulshat.atanbaeva.76@mail.ru (G.K.A.); kulbaevamarzhan931@gmail.com (M.S.K.)

**Keywords:** abdominal cavity, inflammation, lymph nodes, paracortex, cerebral sinus

## Abstract

**Simple Summary:**

Given the important role played by the lymphatic system in tissue drainage, metabolism, water–salt metabolism, and its protective compensatory and immune functions, it is important to further study the role of the lymphatic system in the development of abdominal inflammation. The features of lymph transport through vessels and nodes, the state of formation and biochemical composition of the lymph, the state of innervation of lymph vessels and nodes, and the state of the cellular composition of lymph nodes in the experimental modeling of inflammation in the abdominal cavities of internal organs reveal new insights regarding the mechanisms of this disease. Since, in the literature, only isolated studies regarding the lymphatic system in violation of the abdominal cavity exist, it is clear that the roles and functional states of lymphatic vessels and nodes in inflammation in the abdominal cavity have not yet been studied in a complex and multi-faceted way.

**Abstract:**

This paper provides data regarding the ratios of the functional structures of lymph nodes after inflammation in the abdominal organs. Lymph systems, including lymph nodes, function as filters of tissues and tissue fluids and are places of origin and lymphocyte production for normal physiological functions. They display specific morphological and functional responses in reaction to endogenous and exogenous substances. The morphological pattern of the mesenteric lymph node in experimental rat groups reflects a decrease in its immune function due to the processes of inflammation in the abdominal cavity. These processes work together with the associated organs and their involvement in the abdominal lymph nodes, in which there are discharges of the structure of the paracortical zone under conditions of reduced lymphogenic processes, according to the decrease in the size of the paracortex and the ratios of lymphoid nodes with and without germinal centers. Histological and morphometric analyses show changes in the mesenteric lymph node. These analyses are characterized by changes in the cortical and medullary substances, while the proportion of the cortical structure decreases. We also noted an increase in the number of macrophages in the lymphoid nodes and cerebral sinus, as well as a decrease in the number of mature plasmocytes, the paracortex, and the pulp strands. These changes indicate immunosuppressive effects on the lymph node. Under the conditions of inflammation, the formation of a mixed immune response occurs.

## 1. Introduction

One of the most pressing public health problems is inflammation of the abdominal organs because the number of patients, including those with severe forms of this disease, is constantly increasing. Inflammation in the parietal and visceral layers of the peritoneum is accompanied by severe general health problems in patients [1,2,3].

Internal inflammation causes suppuration, the appearance of excess free fluid in the abdominal cavity, and general body intoxication when it enters the abdominal cavity and affects the gastrointestinal or intestinal contents and microflora, as well as the microbes and bacteria that live in the lumen of the gastrointestinal tract. For acute surgical diseases and abdominal injuries, acute inflammation in the abdominal cavity plays a leading causative role, surpassing widespread peritonitis in terms of severity and fatality. According to past authors, only in our country does the number of patients with acute surgical disorders of the abdominal cavity experience an annual increase, and 25–30% of these diseases are worsened by various types of peritonitis [4,5,6]. Past authors claim that extensive purulent peritonitis is fatal in 20–30% of cases, postoperative peritonitis is fatal in 40–50% of cases, and “tertiary” peritonitis is fatal in 75% of cases [7,8,9].

The risk of lethality is quite significant in cases of severe abdominal cavity inflammation and the growth of the number of affected organs. The intestine is one of the most important organs to primarily experience pathological changes. Mild and severe ischemia cases are distinguished based on the nature of the morphological changes in the intestinal wall, with the necrosis of individual enterocytes in the area of the small intestine’s villi being a hallmark of severe disease [10,11,12].

The mucous membrane of the digestive tube is one of the most significant surfaces of the organism, and it experiences constant interactions with the external environment and warrants further consideration [13,14,15]. The inflammatory process in the abdominal cavity with peritonitis, accompanied by the translocation of bacteria and toxins from the intestine, results in a complex combination of hemodynamic, metabolic, and immune disorders [16,17].

The vascular system is most immediately involved in the pathological process during inflammation in the abdominal organs, as the inflammatory process develops and exudation increases, leading to the expansion of the lymphatic and blood vessels [18,19,20]. The activation of lipid peroxidation processes and the development of endotoxicosis, which are both determinants of critical state formation, are two key mechanisms involved in metabolic disorders resulting from the violation of internal organs [21,22,23]. The formation of hypoxia and, ultimately, the insufficiency of the main cellular energy-generating system, namely mitochondrial oxidative phosphorylation, are recognized as key processes involved in free-radical oxidation violations [24,25,26].

Researchers must recognize the value of investigating lymph nodes in abdominal disorders to better understand the mechanisms driving interactions between inflammatory processes and lymphatic (immune) systems. The lymphatic system provides a barrier against pathogenic effects and is actively involved in the inflammatory process’ pathogenesis [27,28,29]. Increases in the volume density of the hemomicrocirculatory loop and lymphatic sinuses, the inhibition of lymph node immunological activity, and the reduction in the intensity of transendothelial mass transfer in the endotheliocytes of the lymphocytic sinuses are all effects of the inflammation, including disorders affecting internal organs [30,31,32].

Lymph nodes have great functional importance for the body, both in normal and pathological processes [33,34,35]. The lymph node plays a crucial role in the pathogenesis of inflammation [36,37,38]. The lymph nodes experience various protective and adaptive changes, as well as pathogenic changes, because they are the main barrier against the spread of infection from the pathological focus to the bloodstream. Pathogenetic therapy’s ultimate goal of stimulating the body’s natural defenses is achieved by evaluating the lymph nodes’ responses to infectious processes, particularly in light of the growing administration of endolymphatic medication in clinical practice [39,40,41,42].

This investigation will study the morphofunctional state and structural and functional zones of mesenteric lymph nodes under normal conditions and in an experimental setting when the abdominal organs are inflamed.

## 2. Materials and Methods

### 2.1. Ethics Statement

All groups of animals were kept under the same feeding and keeping conditions in the vivarium. All experiments performed on animals were conducted in strict accordance with the rules developed and approved by the Local Ethics Committee of the Institute of Genetics and Physiology, Protocol No. 12-314, of 11 November 2022, as well as the rules of bioethics approved by the European Convention for the Protection of Vertebrates (Strasbourg, 1986) and the guidelines outlined in the European Union Directive 2010/63/EU of 22 September 2010, titled “On the protection of animals used for scientific purposes”.

### 2.2. Animals and Experiment Design

The experiments were conducted on 60 white laboratory male Spraque–Dawley (SD) rats weighing 250 ± 5 g. Three groups of rats were created: one group contained fifteen control rats, and the other two groups contained rats subjected to experimentally induced acute abdominal organ inflammation. The second group (22 rats) experienced inflammation on the second day, while the third group (23 rats) experienced abdominal organ inflammation on the fifth day.

We chose a method for modeling inflammation in the abdominal organs by introducing fecal suspension, which is similar to an acute inflammatory process and represents the completion of the acute phase of peritonitis in terms of etiopathogenesis, clinical manifestation, and phasic flow similar to those in humans. We caused acute inflammation of the abdominal organs in the rats by introducing fecal suspension into the abdominal cavity at a rate of 0.5 mL of a 10% solution per 100 g of animal body weight [43]. No later than 20 min after the preparation stage, the resultant fecal suspension was injected into the abdominal cavity of the animals using the puncture method. Animals’ abdominal cavities were filled with fecal suspension, while the needle tip was vertically positioned, with the caudal being placed end-up to avoid injuring internal organs. The rats underwent abdominal dissections under ether anesthesia. The objects of study were the mesenteric lymph nodes, which were removed for morphological examination. The assessment of the structural and functional zones of the lymph nodes was carried out using histological, cytological, and morphometric methods.

### 2.3. Morphological Research

The lymph nodes were preserved in 10% neutral formalin for histological evaluation using light microscopy. The materials were cleaned in xylene before being histologically examined in increasing concentrations of alcohol, and they were then embedded in paraffin. Using a ThermoScientificHM 325 microtome, 4–5 μm thick histological sections of the lymph nodes were obtained. Histological sections of lymph nodes were stained with hematoxylin–eosin, azure, and eosin and embedded in polystyrene [44,45]. Digital images of histological preparations were obtained using the LEICA DM 750 microscope coupled with a camera and LEICA Application Suite software. In total, 105 preparations of mesenteric lymph nodes were examined.

### 2.4. Morphometric Analysis

Morphometric analysis was performed using a morphometric grid [46,47], which was applied to a section of each lymph node. The longitudinal and transverse dimensions of the lymph nodes were measured. The nodal points on the grid that fell onto each lymph node structure were counted using stereometric principles and the method selected for imposing dotted morphometric grids. For the entire portion of the lymph node, as well as for each of its components, namely the capsule, cortical plateau, lymphoid nodules, paracortex, pulp strands, and sinuses, mesh intersections were tallied and recalculated as percentages [48,49]. In the cytological picture of the lymph node, according to the International Histological Nomenclature, reticular cells that form the skeleton of the node, lymphopoietic cells, such as blasts; medium- and small-sized lymphocytes; plasmocytes, of which there were free macrophages; and a few neutrophils, eosinophils, other cells of the lymphoid tissue, cells performing supporting and phagocytic functions, and peripheral blood cells, were observed. During the cytoanalysis of the structures of lymph nodes and plaques, the number of cells per standard area of 1600 µm^2^ was counted by considering their differentiation into blasts, medium- and small-sized lymphocytes, plasma cells, macrophages, etc. The identifying signs were the size of the cell and its nucleus, the width and color of the cytoplasmic rim, the distribution of chromatin and nucleoli in the nucleus, the location of the nucleus, karyometry, etc. Morphometric analysis of the structural components of the lymph nodes with regard to their specific areas was carried out using a morphometric grid. Then, the number of nodes or intersections on the grid that fell on the entire slice and each structural component were calculated and converted into a percentage. The calculations accounted for the fact that the specific cross-sectional areas of the objects on the cut area corresponded to the specific volume of this object in the sample according to the fundamental Cavalieri–Aker–Glagolev principle of stereology. In the structural and functional zones of the lymph nodes, we counted the number of lymphopoietic cells, such as blasts, medium- and small-sized lymphocytes, plasma cells, macrophages (histiocytes), etc. We studied the lymph node structure in accordance with the requirements for histologic examination, taking into account the refined description scheme. Morphometric analysis of the structural components of the lymph node was performed using a morphometric grid of random pitch, overlaid on the lymph node slice. We counted the number of nodes or grid intersections on the whole slice and separately on each of the following structural components: capsule, cortical plateau, lymphoid nodules (follicles), paracortex, fleshy tracts, and sinuses. When digital data were obtained, their standardization was performed using the matrix statistical method. The matrix method uses the operation “normalization of features” according to the following formula: Np = (Xp—Xk)/Sd, where Np = normalized value; Xp = actual value; Xk = arithmetic mean; and Sd = standard deviation of each used indicator. The normalized indicators are standardized with a (+) or (−) sign and show the deviation from the average value of the given indicator within the limits of ±1.0. For each structural and functional element of the organ, the normalized value was calculated using the subsequent calculation of the total normalized index. At the light-optical level, the lymphoblast and plasmoblast were indistinguishable, and their definitions correlated with the location in the lymph node. Therefore, in the cerebral strands, blasts were designated as plasmoblasts among plasmocytes [50,51]. Taking the cut area to be 100%, we found the relative areas of the capsule, marginal sinuses, lymphoid nodules with reproduction centers, internodular zone, deep zone of the cortex, pulp strands, and cerebral sinuses. The ratio of the specific area of the cortex to the specific area of the medulla (cortical/medullary ratio—C/M) was calculated for the lymph nodes present in each experimental group. The separation of structural components and differentiation of cellular forms in lymph nodes was carried out by taking into account the international histological nomenclature [52,53]. Cells were counted in sections in 4 fields of view: the internodular zone, reproduction centers, pulp strands, and cerebral sinuses. Reference values for the cells were used for some structural and functional zones.

### 2.5. Statistical Analysis

Statistical processing of the obtained results was carried out using the StatPlus Pro 2009 program (AnalystSoft, Inc., Alexandria, VA, USA) using Student’s *t*-test. The arithmetic mean (*M*) ± and mean error (±*m*) were used to present the data. Differences were considered to be significant at *p* < 0.05.

## 3. Results

In the morphological study, the mesenteric lymph nodes of the control animals were covered by a distinct thin capsule, and the marginal sinus was not identified. The mesenteric lymph nodes of the rat are located in the area extending from the ileocecal corner of the intestine to the root of the mesentery. The shapes and sizes of these nodes differ but oval, bean-shaped, and horseshoe-shaped dimensions are common. Lymphoid nodules are clearly expressed in the lymph nodes. Lymphoid nodules are predominantly round in shape and located in a single layer in the peripheral zone of the cortical substance, and they have clear contours (Figure 1A). In the early stages of inflammation, a microscopic examination of the mesenteric lymph nodes revealed a significant abundance of tissue, edema, lymphocytic and leukocyte infiltration of the capsule, the stroma of the cortex, the medulla of the node, and extended subcapsular sinuses. Mesenteric lymph node edema was observed in certain animals. Intestinal wall edema and enlargement of the mesenteric lymph nodes were more pronounced than on the second day of the experiment. The sinuses were enlarged, the erythrocytes were damaged, and many cells loaded with hemosiderin (siderophages) were identified in their lumen in rats with abdominal cavity lymph node inflammation. The presence of siderophages indicates the increased destruction of erythrocytes due to the activation of lipid peroxidation.

On the fifth day of the experiment, the lymph nodes became even more full blooded, and the venules of the paracortex and the medulla of the lymph node were dilated. The expansion of the sinuses of the lymph nodes and swelling of reticuloendothelial cells were noted. In the cavities of the sinuses of reticular cells, lymphocytes and macrophages were recorded. An increase in the sizes of the lymphoid follicles was associated with increases in their germinal centers, in which plasmablasts and plasmocytes appear. Plasma cells were also registered as part of the brain cords. The reticuloendothelial cells of the sinuses reorganize into macrophages. The paracortical zone was reduced 2 times compared to the previous period of observation, and the hyperhydration of the parenchymata of the lymph nodes increased (Figure 1B). Follicular hyperplasia was observed in lymph nodes, characterized by the enlargement of follicles and expansion of reproduction centers. Often, these changes were combined with plasmocytes in the medullary tracts and interfollicular parenchymata. In the lymphatic follicles, we also observed the pronounced desquamation of the reticular stroma of the germinative centers of follicles (Figure 1C).

The structures of the mesenteric lymph nodes altered their typical structures, the clear boundary between the cortical and medulla disappeared, and the number and sizes of the lymphoid nodules changed. The areas of the lymphoid nodules without a germinal center somewhat increased, and the area of lymphoid nodules with a germinal center slightly decreased on the second day of inflammation in the abdominal organs. The areas of the lymphoid nodules gradually diminished on the fifth day of peritonitis, and their germinal centers were essentially undefinable or showed signs of atrophy. Attenuated lymphoid nodules were found in the cortex and were a morphological manifestation of a decrease in immune defense (Table 1).

This study showed 1.36- and 1.43-time decreases in the paracortex in the lymph nodes compared with the control group (Table 1). As inflammation in the abdominal cavity progressed, the severity of both hemodynamic and reactive changes in the lymphoid tissue became aggravated. Changes in the size of lymphoid nodules occurred.

The main structural and functional zones of the rat mesenteric lymph nodes underwent modifications that point to their minimization with the lymph node’s compaction, which is accompanied by a reduction in the overall area. The cortical brain index decreased from 0.51 to 0.49. We also noted a decrease in the immune potential of both humoral and cellular aspects (Table 1). The dynamics of the number of lymphoid nodules with a light center and their linear dimensions, as well as changes in the structure of the medullas of the lymph nodes, suggested a change in their immune function.

Consequently, the specific size of the interfollicular zone in the mesenteric lymph nodes increased as the inflammatory processes in the abdominal organs developed. With regard to morphometric measures, a statistically significant decrease in the paracortical zone’s percentage (by 30.66%, *p* < 0.001) compared to the control group was observed; however, this change was followed by a rise in the average lymphocyte count (*p* < 0.001). The relative area of the cerebral cords decreased by 29.64% (*p* < 0.001), indicative of the activation of blast transformation processes in this region of the mesenteric lymph nodes. However, this decrease was accompanied by a statistically significant increase in the number of medium lymphocytes, plasmablasts, immature plasma cells, and macrophages (Table 1).

The majority of the mesenteric lymph nodes in the control rats were represented by cortical substances on histological sections, and the border between the cortical and medulla was visible. The paracortical zone and the parenchyma of the lymph node between the lymph nodes made up the cortical plateau, which, together with the cortical substance of the lymph nodes, constituted the T-zone-dependent area of the lymph node. The germinal center and the mantle zone around it, which made up the lymph node’s central light part, were the B-zone-dependent areas of the lymph node. The main part of the medulla was formed by clearly distinguishable pulp strands oriented in the direction of the gate. Low magnification microscopy revealed that brain cables, sinuses, and follicles with paracortical zones reflected the lymph nodes’ cortices and medullas. We found it challenging to discern between the T- and B-zones in primary follicles with monomorphic cellular compositions (Figure 2A).

Follicular hyperplasia, characterized by an increase in follicles and the expansion of reproduction centers, is observed in the lymph nodes of animals experiencing inflammation in the abdomen cavity. These alterations frequently occur in conjunction with the development of plasma cells in the brain cords and interfollicular parenchyma. Little lymphocytes make up the unique mantle zone remaining within the follicles, and the polarization of centroblasts and centrocytes in the reproductive centers can be identified. They are mostly found in the node’s cortex and frequently differ in size and structure. Many apoptotic bodies and mitotic figures are observed in reactive breeding facilities; these entities are frequently phagocytosed by “stained body macrophages” (Figure 2B).

The cortical zone of the mesenteric lymph node is pushed to the periphery of the organ; however, it has well-defined primary and single secondary lymph nodes. The cortical plateau then narrows; however, it is represented by a fairly dense population of cellular elements (Figure 2B). The paracortical zone of the lymph node is represented by rather densely located small- and medium-sized lymphocytes, phagocytically active cells, and reticular cells. A significant section of the paracortical zone is occupied by dilated intermediate sinuses. Dilated vessels of the microvasculature are also noted, often in tandem with perivascular lymphoid infiltration.

The paracortical zone has an uneven width, and its noticeable expansion is linked to changes in the distributions of the cells’ densities, particularly at its boundary with the medulla (with pulp strands). Depending on whether they belong to the humoral (B-zone-dependent) or cellular (T-zone-dependent) immune systems, the lymphoid series of the parenchymata of the lymph nodes can be extremely susceptible to outside influences, which cause changes in the structural and functional zones of the lymph nodes.

They are usually predominantly distributed in the cortex and often vary in shape and size. The relative cellular composition of the cortical plateau (interstitial zone of the cortical substance) of the lymph node is characterized by the predominance of small lymphocytes located between the processes of the reticular cells. Single macrophages, cells of the plasmacytic series, and immature cells of the lymphoid series are observed. Changes in the relative cellular composition of the plateau cortex in postnatal ontogenesis are characterized by a general decrease in the density of lymphoid cells. Plasmacytoid monocytes may occur singly, in small clusters, or in large aggregations that mimic follicular centers. Although these cells’ proliferative activity is low, the detection of apoptotic cells in groups of plasmacytoid monocytes is typical (Figure 3).

The medulla fills the lymph node’s center and is situated closer to the lymph node’s hilum in histological preparations. It is also lighter in color. In several cases, the medulla expands the cortical substance and, in narrow areas, penetrates the peripheral sections of the cortical plateau and the paracortical zone. The parenchyma of the medulla is represented by medullary (pulp) strands (Figure 4).

The cerebral sinus and pulp strand cytoarchitecture of the lymph node’s medulla change. As the number of lymphocytes increases (by 1.7–3.0 times), the number of plasmocytes (by 3 times), reticular cells (by 1.8 times), and pulp threads decrease. We also noted a decrease in the formation of plasma cells in the pulp strands in line with the number of blasts within the control limits under experimental conditions. In the cerebral sinus nodes, increases in the number of lymphocytes (by 1.92 times) and macrophages (by 2.44 times) and a decrease in the number of plasma cells (by 1.92 times) were noted. In all probability, some of the cellular elements of the mesenteric nodes died, especially in the lymphoid tissue (Figure 4).

## 4. Discussion

The animals’ motor activity decreased 48 h after simulating the experimental inflammation of the abdominal organs. The caged rats were lethargic but walked freely. They consumed less food, and frequent, loose stools were produced. The animals’ appearances were characterized by thin, brittle hair and increased hair loss. Animal mortality after 24 and 60 h was as follows: in the second group, 25% of the rats died, and in the third group, 35% of the rats died, i.e., seven rats. Morphological changes in the abdominal cavity of rats were characterized using a visual descriptive method. There were no significant deviations in either the experimental group or the control group in the abdominal cavity of the stomach wall.

An autopsy of the abdominal cavity in animals on the fifth day revealed 1.5–3.0 mL of a turbid liquid with a pungent odor; some animals had a purulent effusion with an unpleasant odor. Upon opening the abdominal cavity, the loops of the small and large intestines were slightly inflated, and peristalsis was preserved. The peritoneum and serosa of the intestine were dull; the omentum and serous membranes of the internal organs of the abdominal cavity were hyperemic, with a coating of fibrin; and the intestinal loops were paretic and swollen throughout. The data obtained showed that the morphological changes in inflammation in the abdominal cavity reflected the characteristics of peritonitis. The abdominal cavity disorder, as well as disorders related to blood circulation and the small intestine, manifested in the form of a plethora, and, in some places, vascular thrombosis occurred. Numerous aerobic and anaerobic microorganisms were discovered during acute inflammation in the abdominal cavities of rats. Additionally, lympho- and hemodynamics, as well as the physicochemical parameters of lymph and blood, were violated [54,55], and the absence of peristalsis eliminated intestinal colonization resistance, enabled the translocation of pathogenic and opportunistic pathogenic microflora into habitats that were unusual for it, and led to bacteremia, the development of abdominal sepsis, and multiple organ failure [56]. All these facts are evidence of the profound changes in the blood and lymph nodes during experimental fecal inflammation in the abdominal organs.

The marginal sinus was not identified during the morphological study of the mesenteric lymph nodes in the control animals, which were covered by a distinct thin capsule. Rats’ mesenteric lymph nodes are located in the area extending from the ileocecal corner of the intestine to the root of the mesentery. The shape and size of these nodes differ in variety; typical dimensions include oval, bean-shaped, and horseshoe-shaped nodes. Lymphoid nodules are clearly expressed in the lymph nodes. Lymphoid nodules are predominantly round in shape and located in a single layer in the peripheral zone of the cortical substance, and they have clear contours. During the micro-preparations of the mesenteric lymph node, the number of lymphoid nodules with centers was 1.72 ± 0.2, and without a germinal center, the node occupies 1.33 ± 0.11 microns of the node area. The lymph nodes’ paracortices were concentrated via the accumulation of follicles and had discrete formations with blurry outlines without distinct boundaries. The cortical layer was represented by many lymphatic follicles, the bulk of which were follicles without light centers. Follicles with light centers were rare. The medulla of the node had a looser structure, revealing connective tissue trabeculae, cerebral sinuses, reticular tissue, and vessels. Most often, small lymphocytes were found in the lymphoid tissue of the cortex and medulla, while medium lymphocytes were found in the light centers. A large number of reticular cells were found in the cerebral sinuses. This is due to their higher functional activity, the intensity of immunological processes, and the regional features of the inflowing lymph [57,58,59].

The mesenteric lymph nodes’ microscopy results during the studied periods of inflammation in the abdominal cavity indicated the progress of destructive changes (the appearance of cells in a state of destruction) against a background of the development of inflammatory processes in the mesenteric lymph nodes, and there were circulatory disorders in the mesenteric region, and with the foci of hemorrhage, a sharp expansion of venous vessels occurred. We concluded that the inflammation of the abdominal organs is accompanied by structural changes in the lymph nodes, inhibiting the function of these organs, expanding the sinuses, and causing the uneven thickening of the argyrophilic fibers.

The mesenteric lymph nodes of rats in the control group were investigated. The mesenteric lymph nodes were covered with a thin, dense capsule. The subcapsular (marginal) sinus was prominent and heavily cellularized. We noted a clear cortical–brain border in the mesenteric nodes, and the C/B index was 0.84. The cortical substance’s area noticeably predominated over the cortical substance. Lymphoid nodules were present in the cortex, including more nodules without a germinal center. The lymphoid nodules ranged in size. The paracortex had an uneven width under the lymphoid nodules and was well vascularized. At the experiment’s end, the lymphoblast counts in the lymphoid nodules had decreased 1.85 times, while the average lymphocyte counts had increased 2.5 times, and the macrophage counts had increased 2.1 times. These changes may result from the reduction in the abdominal organs’ inflammation progress. As previously mentioned, the ratio of different structural and functional zones of the lymph nodes changed under the new conditions of flight and the impact of weightlessness on the body of mice.

The reduction in the thymus-dependent areas (T-zone) also confirmed this belief. Compared to the control, the area of the paracortex was smaller during abdominal cavity inflammation, which worsened the pathology’s prognosis. Maintaining or expanding the T-zones was more frequently linked to a positive prognosis [60,61,62,63]. The B-dependent zone dominated the node’s structure, indicating that humoral immunological activities predominated in that area. The widths of the medullary cords of the medulla were initially lower by the second day, before increasing by the fifth day of inflammation in the abdominal organs.

The ratio of lymphoid nodules with and without a germinal center remained quite high in the structural and functional zones of the mesenteric lymph nodes, indicating both the predominance of lymphoid nodules with germinal centers and the active differentiation of cells in the lymph nodes. The overall areas of the lymph nodes tended to grow once the observed abdominal organs became inflamed, along with a shift to a compact morphotype where the cortical substance predominated. The cortical–cerebral index value, which is 0.84 ± 0.13 (with inflammatory processes 0.51 ± 0.16), proved this point. In the structure of the lymph node, we noted 15–20% and 264–221% increases in the areas of the capsule and the subcapsular sinus, respectively.

Secondary follicles were common and mainly located closer to the central parts of the lymph nodes. The most obvious germinal centers of lymphatic nodules were mainly determined in the control group. Large light immunoblasts with surrounding microenvironment cells and solitary macrophages were also present in the germinal centers of the lymphatic nodules. Plasmacytoid monocytes, which can be observed individually, in small groups, or in huge clusters that resemble follicular centers, were present in the paracortical zone at different densities. The cells had identical sizes, were round, and often had eccentric nuclei, in which there was no condensed chromatin characteristic of plasma cells. Even though these cells achieved only modest levels of proliferative activity, it was typical to find apoptotic cells in collections of plasmacytoid monocytes. The cords included a varied mixture of tiny lymphocytes, blast cells, and plasma cells, and they were directed toward their lymph nodes’ hila.

The cellular composition of the T-dependent paracortical zone of the mesenteric lymph nodes was characterized by the predominance of mature cells of the lymphoid series (small lymphocytes) and was located between the microenvironment cells. Plasmacytic series cells were observed individually due to the functional characteristics of the paracortical zone of the lymph node.

We believe that under inflammatory conditions in the abdominal organs in rats, a weakening of humoral immunity occurs in line with the preservation of the immune response of the cellular type. Noticeable expansions of the paracortical zone and sinuses of the nodes are morphological indicators of increases in the functional activity of the T-dependent zone and the transit function of the sinuses in the lymph nodes. The cortical plateau of the nodes in the experimental animal group reduces following inflammation due to an increase in the lymphoid nodule-occupied region of the cerebral sinuses. We noted a 1.26-fold (26%) reduction in the size of each lymphoid nodule with a germinal core. We also noted a 1.29-time decrease in the area of the paracortex (decreases of 26.7% and 30.3%, respectively).

According to our observations, the mesenteric lymph nodes of rats had a weakly fragmented cortical substance. In all probability, follicular hyperplasia after the occurrence of inflammation in the abdominal organs of the body is the most common type of lymph node reaction and is characterized by increases in follicles and the expansion of the reproduction centers. This type of lymph node is known to be capable of performing its functions effectively [64,65], and lymph nodes play important roles as active barriers to prevent the spread of pathogens along the lymphatic tract [66].

Follicular hyperplasia and isolated follicles with destroyed light centers were observed in the control group. Medium and small lymphocytes, macrophages, and plasma cells are the main cell types found in the cerebral sinuses [67]. The quantity of medium lymphocytes changes during inflammatory events in the mesenteric lymph nodes. In the examined lymph nodes’ cerebral sinuses, macrophages, plasma, and destructively altered cell development are measured and compared to intact values recorded on the second and fifth days following abdominal cavity inflammation. The capsule thickness of the lymph nodes under study is likewise marginally thicker than that of the control, suggesting a small reduction in lymph node size.

As a regulator of immunological and water tissue homeostasis, the lymphatic system is essential for pathogenesis and sanogenesis [68,69,70,71]. It reflects the morphofunctional state of the lymph nodes, which varies as a result of the inflammatory processes taking place in the lymphatic region of the organ. With the entry of fecal waste into an animal’s body, a change in the functional activity of the abdominal organs is noted, impacting the shape and function of the lymphoid nodules (follicles) in the lymph nodes, and the lymphopoietic function is diminished in inflammatory processes.

Thus, in accordance with the experimental investigations, we can state that the morphological and functional changes that occur in the mesenteric lymph nodes sufficiently reflect the dynamics of the inflammatory process that occurs with induced fecal suspension in experimental animals. The results of the studies showed that in the case of inflammation in the abdominal organs, compared with the control, the areas of lymphoid nodules with a germinal center, the cerebral sinus, the areas of the paracortex and the pulp strands, and lymphoid nodules without a germinal center increased. For the inflammatory processes of the abdominal organs, individual follicles with small light centers remained, and most of the follicles had primary (unstimulated) structures without reactive centers. The areas of the paracortical zone and sinuses decreased; however, the paracortical zone was still weakly expressed, indicating a continuous deficit of T-lymphocytes, against the background of the active proliferation of B-lymphocytes in the germinal center of the lymphoid follicles. The latter were sometimes slightly dilated and mainly contained separate reticular cells connected through long cytoplasmic processes. For the cortical plateau and pulp strands, an insignificant number of plasma cells was found. On the second day after inflammation, the pulp strand study revealed a different pattern, with a 1.13-time decrease recorded, and a 1.24-time increase was recorded on the fifth day of abdominal organ inflammation (showing a multidirectional character on the second day, with a decrease of 12.5%, and on the fifth day, with an increase of 24.6% recorded). The number of cerebral sinuses decreased on the fifth day of abdominal organ inflammation but was still 29.2% larger than that of the control group. In the cerebral sinus, we recorded increases of 74% and 29.2%, respectively.

## 5. Conclusions

Our results highlighted the dependence of the structures of the lymph nodes on the functional states of the abdominal organs. Compared to the experimental group, the lymph nodes in the control animals displayed inflammatory reactions, attributed to the lymph nodes’ morphofunctional statuses and structural changes that occurred during inflammatory processes in the abdominal organs. The changes noted in the lymph nodes of the experimental group animals are less pronounced in the transformation of the structural and functional zones. We observed that the sinuses of the lymph nodes had widened and the reticuloendothelial cells had swollen. Lymphocytes and macrophages were found in the sinus cavities of reticular cells. The lymph nodes became even more full blooded after 2 to 5 days of the experiment. Increases in the sizes of the lymphoid follicles are associated with increases in their germinal centers, in which plasmablasts and plasmocytes appear. Plasma cells are also registered as parts of the brain. The reticuloendothelial cells of the sinuses are reorganized into macrophages. Compared to the control group, the paracortical zone underwent a 30.66% reduction. Under conditions of inflammation in the abdominal cavity, the lymph node was compacted, decreasing the main structural and functional zones. This indirectly indicates a regional deficiency of cellular and humoral immunity and can serve as a marker of this condition.

## Figures and Tables

**Figure 1 biology-13-00166-f001:**
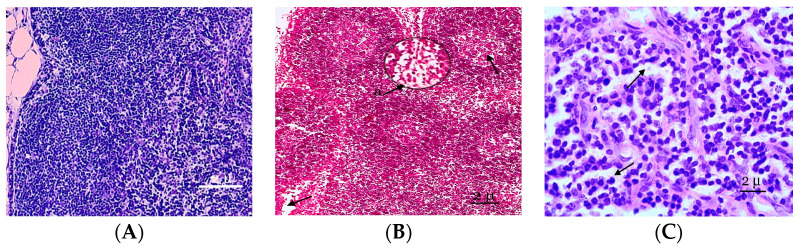
A fragment of the cortical substance of the mesenteric lymph node. (**A**) Mesenteric lymph nodes of the control group of rats studied at the histological level are represented by well-defined structural and functional zones. (**B**) Fragment of the cortical substance of the mesenteric lymph node. Fifth day of the experiment. a—A reduction in the paracortical zone. (**C**) The pronounced desolation of the reticular stroma of the germinal centers of the follicles. Fixation: 10% neutral formalin solution. Staining: Mayer’s hematoxylin and eosin. Zoom: ×200. Showing the paracortical zone was reduced and desquamation of the reticular stroma (with arrows).

**Figure 2 biology-13-00166-f002:**
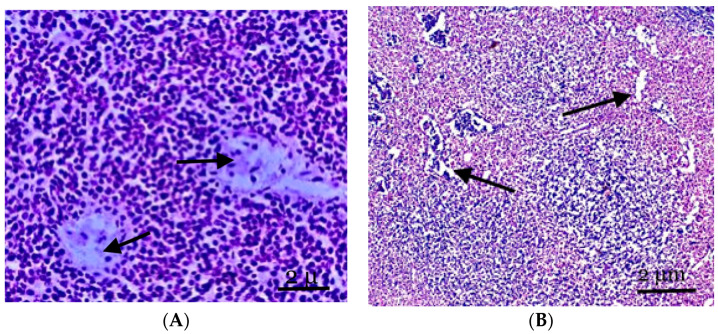
(**A**) The structure of the lymph nodes of the control animals. The cortical substance is represented by cellular elements, mainly lymphocytes against the background of two central arteries and part of the paracortical zone. (**B**) The mesenteric lymph node of the experimental group of animals. The expansion of the vessels of the microvasculature, sinus spaces, and pronounced lymphoid nodules, with the cortical substance displaced to the periphery of the node. Staining with hematoxylin and eosin was performed. Zoom: ×200. Showing the part of the paracortical zone and the expansion of the vessels of the microvasculature, sinus spaces (with arrows).

**Figure 3 biology-13-00166-f003:**
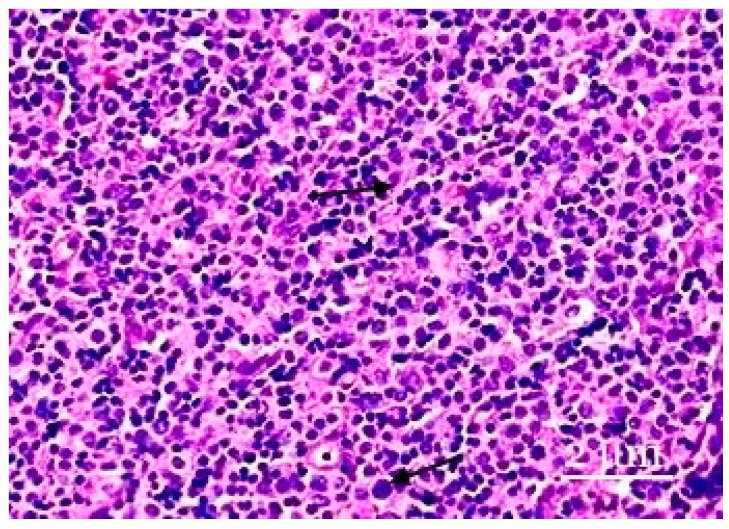
The paracortical zone of the lymph nodes. The paracortical zone has a more pronounced inflammatory response, predominantly caused by neutrophils. Plasmacytoid monocytes are in the form of either small groups or large clusters. Hematoxylin–eosin staining was performed. Zoom: ×200. Showing the immature cells and plasmacytoid monocytes (with arrows).

**Figure 4 biology-13-00166-f004:**
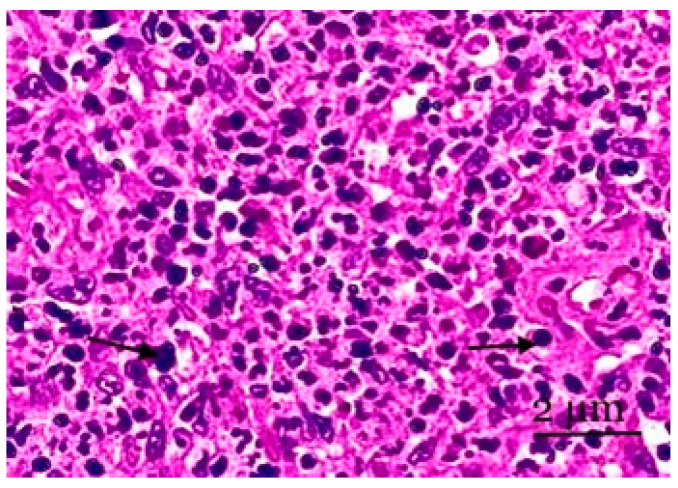
A section of the medulla of the mesenteric lymph node. Lymphocytes of various sizes and plasma cells are observed in the cerebral sinus. Cortical pathways have the sinuses of the medulla located between them. Staining with hematoxylin and eosin was performed. Zoom: ×200. Showing the lymphocytes of various sizes and plasma cells (with arrows).

**Table 1 biology-13-00166-t001:** The characteristics of the structural and functional zones of the mesenteric lymph nodes in the control group and after the experimental inflammation of the abdominal organs, µm^2^.

Structures of the Mesenteric Lymph Nodes	Control	Two Days after Inflammation	Five Days after Inflammation
	1	2	3
Capsule	0.98 ± 0.07	1.13 ± 0.08	1.18 ± 0.04
Subcapsular sinus	0.14 ± 0.04	0.51 ± 0.09 **	0.45 ± 0.06 **
Cortical plateau	1.06 ± 0.07	0.70 ± 0.04 *	0.71 ± 0.08 *
Lymphoid nodule without a germinal center (F1)	1.33 ± 0.011	1.72 ± 0.05	1.68 ± 0.13 *•
Lymphoid nodule with germinal center (F2)	1.72 ± 0.02	1.02 ± 0.07 *	0.93 ± 0.02 *•
Paracortex	4.46 ± 0.35	3.27 ± 0.42 *	3.11 ± 0.21 *
Pulp strands	5.41 ± 0.12	4.78 ± 0.08 *	6.74 ± 0.18 *•
Cerebral sinus	2.5 ± 0.11	4.35 ± 0.25 *	3.23 ± 0.25 *•
Total area	17.58 ± 1.99	19.51 ± 2.11	17.98 ± 2.12
Cortical brain index	0.84 ± 0.013	0.51 ±0.16	0.49 ± 0.20
F2/F1	1.29 ± 0.012	0.59 ± 0.04	0.58 ± 0.06
T-zone	5.52 ± 0.21	3.97 ± 0.23 *	3.89 ± 0.29 *
B-zone	8.46 ± 0.35	7.51 ± 0.38	8.52 ± 0.24
Index T/B	0.65 ± 0.26	0.53 ± 0.29	0.46 ± 0.26

Note: * *p*_1–2,3_ < 0.05; ** *p*_1–2,3_ < 0.01; • *p*_2–3_ < 0.05—reliability of differences between indicators.

## Data Availability

The data presented in this study are available on request from the corresponding author.

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
