# Peer review of "Morphofunctional Alteration of Mesenteric Lymph Nodes in the Inflammation of the Abdominal Cavity"

_biology, 2024, doi:10.3390/biology13030166_

Round 1
Reviewer 1 Report (Previous Reviewer 1)
Comments and Suggestions for Authors
The authors have made essential modifications and as a result, the manuscript's quality has improved. I wish the authors the best of luck.
Author Response
Dear Reviewer let me thank you for your hard work in reviewing our article. Thank you very much again. Thank you.
Sincerely, author Serik Abdreshov

Reviewer 2 Report (Previous Reviewer 2)
Comments and Suggestions for Authors
The resubmitted manuscript by Abdreshov and colleagues does not address my main concerns. The provided images do not prove the morphological alterations of mesenteric lymph nodes during inflammation. Additionally, it is not clear how the authors generated the data presented in Table 1.
The aim of the study is interesting but the authors need a range of microscopic images to illustrate the described morphological changes. The present manuscript does not provide enough evidence to support the conclusions.
Comments on the Quality of English LanguageAn improved version that still needs editing, especially in the abstract.
Author Response
Dear Reviewer let me thank you for your hard work in reviewing our article. Thank you very much again. Thank you.
Sincerely, author Serik Abdreshov

Reviewer 3 Report (Previous Reviewer 3)
Comments and Suggestions for Authors
The reviewer is satisfied with the revision, and recommens being published.
Author Response
Dear Reviewer let me thank you for your hard work in reviewing our article. Thank you very much again. Thank you.
Sincerely, author Serik Abdreshov

This manuscript is a resubmission of an earlier submission. The following is a list of the peer review reports and author responses from that submission.
Round 1
Reviewer 1 Report
Comments and Suggestions for Authors
The authors in this manuscript presents data on the functional structural ratios of lymph nodes post abdominal organ inflammation. The structural pattern of mesenteric lymph nodes in experimental rat groups represents a diminished immune function owing to abdominal cavity inflammation, as well as the related organs. This is demonstrated by alterations in the structure of the paracortical zone as well as changes in histological and morphometric studies. These modifications suggest immunosuppressive effects on the lymph node and the development of a mixed immune response in the context of inflammatory events. Here are my comments:
1. The sentence from line 17 to line 21 is too long. Please consider splitting into two sentences at least.
2. Change the word “met” in line 18 and substitute with more appropriate word such as “found or encountered or discovered or noticed or observed’.
3. Line 18 there are two “in”
4. Please explain or rephrase the sentence in line 21 “An article is presented that is part of this research paper”
5. There are many words in this manuscript that lacks space between two words. For example in line 25, spacing is required between the words “including” and “lymph”. The authors have written as “includinglymph”. In fact, there are many such words that needs spacing between two words. Please correct all the non-spaced words in line 28, 30, 36, 38, 45, 49, 55, 56, 58, 59, 60, 61,67, 73, 74, 78, 87, 91, 92, 97, 98, 103, 122, 136, 134, 130, 144, 148, 149, 155, 160, 176, 182, 187, 238, 338,345.
6. Capitalized the initial letter in the word “together” in line 30.
7. Replace the word “involved” in line 30 with “involvement”.
8. Consider shortening the sentence or splitting it into two-three sentences from line 33 to 37.
9. At the end of the abstract please provide a sentence on the objective of this manuscript.
10. Rephrase the sentence in line “136 to 137”. This may help readers easier to understand.
11. Please clarify what the authors are trying to explain “a modest rise in the liver” in line 181. Are you explaining about the edema?
12. In figure 1, a control section is required to compare and draw the conclusion to make the observation clearer and stronger. Either an uninflamed section or a different day time point when there is no substantial difference in edema observed could be utilized.
13. Please put a scale bar in all the figures of the stained sections and label/mark the areas.
14. Figure 1 and Figure 2 could be placed side by side for an easier comparison.
15. Figure 3, please mark the areas/zones in the lymph node.
16. Figure 4, please confirm if the image is a 400X magnification. The cells/tissue of the mesenteric lymph nodes appears tinier than the rest of the 200X images.
17. Figure 3 and Figure 4 could be placed side by side for an easier comparison.
18. Figure 5, please explain, on the basis of the H & E staining how did the authors differentiate monocytes, neutrophils and other inflammatory cells assuming that hematoxylin stains all the nucleus of cells.
19. The discussion section of the manuscript is very detailed and lengthy. Please concise this part. Some of the paragraphs/sentences that broadly discuss about lymph nodes could be omitted. For example lines 330 to 337 can be omitted.
20. Some sentences in the discussion part can be omitted by not addressing the detailed results that were previously examined and explained in the Results section. There are very nice citations that the authors have cited. This sections of the paper might involve explaining more of the current outcomes that the authors noticed and comparing them to those referenced observations.
Comments on the Quality of English LanguageThe English language employed in this work may be improved. Many big sentences may be split down into shorter ones to make it easier for readers to comprehend. Some sentences can be reframed to avoid ambiguity for the readers. I discovered around 40 words that require spacing between two words. I would like to propose that the authors look at this topic further. Some of the terminology may need to be changed to make them more scientifically appropriate.
Author Response
I am very grateful to the reviewers, thank you for the huge amount of work.
Yes, we agree with the reviewer. There are some mistakes and some omissions, we tried to fix everything.
Sincerely, author Serik Abdreshov

Reviewer 2 Report
Comments and Suggestions for Authors
Abdreshov and colleagues used an experimental acute abdominal inflammation (peritonitis) model in rats in order to perform a histologic and morphometric analysis of mesenterial lymph nodes (LNs). The aim of the study is interesting and can shed light to the mechanisms underlying the immune system response in acute inflammation. However, the manuscript has weaknesses that preclude publication at this stage. Specifically:
a) A morphometric analysis relies on data regarding dimensions (size) and weight of LNs. This is not present in the manuscript
b) Histologic analysis should be accompanied by a set of representing images. We cannot assess the morphology of a LN without any low magnification images. The same should be done for primary-secondary lymphoid follicles evaluation, cortex-medulla morphology, sinuses architecture etc. The images in the manuscript are of poor quality and the magnification is too high to access the morphology of the LN. The manuscripy lacks images, sets of images from low to high magnification should be included to demonstrate the authors' findings.
c) It is not clear how the data presented in Table 1 is generated. Additionally, the counting of cell populations (plasma cells, dendritic, lymphocytes cells) should be accompanied by an immunohistochemical analysis that would confirm the hematoxylin-eosin based results.
d) The manuscript is poorly written and needs extensive editing.
Comments on the Quality of English LanguageThe manuscript is poorly written and needs extensive editing. There are many typos and no spaces between words.
Author Response

(The authors gave the same response as above.)

Reviewer 3 Report
Comments and Suggestions for Authors
Please see the attachment.

Author Response

(The authors gave the same response as above.)
